# Computational Evaluation of N-Based Transannular Interactions in Some Model Fused Medium-Sized Heterocyclic Systems and Implications for Drug Design

**DOI:** 10.3390/molecules28041631

**Published:** 2023-02-08

**Authors:** Renate Griffith, John B. Bremner

**Affiliations:** 1School of Natural Sciences (Chemistry), College of Sciences and Engineering, University of Tasmania, Private Bag 75, Hobart, TAS 7001, Australia; 2School of Chemistry and Molecular Bioscience, Faculty of Science, Medicine and Health, University of Wollongong, Wollongong, NSW 2522, Australia

**Keywords:** medium sized heterocycles, protopine, coptisine, transannular interactions, density functional theory, molecular shapes, electrostatic potentials, electron density surfaces, electrostatic charges

## Abstract

As part of a project on fused medium-sized ring systems as potential drugs, we have previously demonstrated the usefulness of Density Functional Theory (DFT) to evaluate amine nitrogen-based transannular interactions across the central 10-membered ring in the bioactive dibenzazecine alkaloid, protopine. A range of related hypothetical systems have been investigated, together with transannular interactions involving ring-embedded imino or azo group nitrogens and atoms or groups (Y) across the ring. Electrostatic potential energies mapped onto electron density surfaces in the different ring conformations were evaluated in order to characterise these conformations. Unexpectedly, the presence of sp^2^ hybridised nitrogen atoms in the medium-sized rings did not influence the conformations appreciably. The strength and type of the N^…^Y interactions are determined primarily by the nature of Y. This is also the case when the substituent on the interacting nitrogen is varied from CH_3_ (protopine) to H or OH. With Y = BOH, very strong interactions were observed in protopine analogues, as well as in rings incorporating imino or azo groups. Strong to moderate interactions were observed with Y = CS, CO and SO in all ring systems. Weaker interactions were observed with Y = S, O and weaker ones again with an sp^3^ hybridised carbon (Y = CH_2_). The transannular interactions can influence conformational preferencing and shape and change electron distributions at key sites, which theoretically could modify properties of the molecules while providing new or enhanced sites for biological target interactions, such as the H or OH substituent. The prediction of new strong transannular interaction types such as with Y = BOH and CS should be helpful in informing priorities for synthesis and other experimental studies.

## 1. Introduction

Ring-fused nitrogen heterocycles form important scaffolds in drug design and development [1,2,3]. Such compounds include many alkaloidal natural products, for example the bioactive protoberberine alkaloids such as berberine (**1**) [4] and coptisine (**4**) [5] and reduced derivatives like the dihydro- (**2** and **5**) [6] or tetrahydro-protoberberines (**3** and **6**) (Figure 1) together with modified protoberberines [7,8] and [9] (pp. 65–68, 95–102). Other related systems include those where the central C-N bond is absent resulting in a core dibenzo-fused azecine skeleton with a central 10-membered heterocyclic system. Allocryptopine (**7**) and protopine (**8**) (Figure 2) are alkaloid examples of this dibenz[*c*,*g*]azecine type. These compounds do not have a quaternary nitrogen site but still display a range of biological activities [10,11,12,13,14], some of which are similar to those exhibited by berberine or coptisine.

Berberine and the closely related coptisine have multifaceted activities [15,16,17]. There seems to be something special about the overall shape and charge distributions in these molecules associated with the fused polycyclic array and nitrogen placement being favourable for efficacious binding interactions with a variety of biological target macromolecules. It was thus of interest from a new drug design point of view to look at whether similar shapes and charge distributions could be accessed through conformational preferencing in related fused medium ring compounds like protopine and model systems derived from it. Potentially more flexibility in design might be achievable through incorporating other atoms (or atom groups) in the medium ring at the transannular interaction site with the possibility then for differential binding to biological target sites or binding to quite different sites. The qualitative and quantitative evaluation of such transannular interactions is thus very important.

A transannular interaction of the (R)N^…^Y type is well known in protopine (**9,** where N is tertiary with a lone pair of electrons, R is a methyl substituent, and Y is a carbonyl group) [18,19]. We have established in previous work the usefulness of Density Functional Theory in predicting this interaction and assessing it in detail in protopine and a few related hypothetical systems [20] with Y = CS and BOH and R = CH_3_ in (**9**). A preliminary investigation also indicated the possibility for an attractive interaction for a number of other Y groups not included in the detailed study. This computational work has now been significantly extended with an emphasis on the influence on conformational preferencing, overall shape and charge distributions. The computational evaluation has also been taken further to include ring-embedded imino (**10**) ([9], pp. 79–81) or azo (**11**) group nitrogens (see Figure 4 below for structures of (**10**) and (**11**)) and atoms or groups Y across the 10-membered ring system, the aim of which was to assess whether some flattening of the ring could be achieved while maintaining additional control of conformational preferencing [21] through transannular N^…^Y interactions. The flattening and lack of a substituent on nitrogen was expected to bring the shape comparison closer to that of the protoberberine alkaloids. The Y atoms or groups considered in this study include C=O, C=S, BOH, S=O, C=CH_2_, C=CF_2_, O, S, and CH_2_. The substituents (R) on the sp^3^ hybridised N were extended from the original Me [20] to H and OH, where the polarization of the N-R bond would be different from N-Me. A number of strong to moderate interactions were indicated in the N-Me (included for comparison purposes), N-H and N-OH systems. In addition, it was shown that imino or azo nitrogens with a free lone pair on nitrogen could also engage in similar strong to moderate interactions with some Y groups. The correlations with shape and electrostatic surface charges were also assessed and the results are presented in this paper, as well as compounds suggested for synthesis and other possible chemical and biological experimental studies.

## 2. Results and Discussion

The DFT-based computations were performed as described earlier using an augmented 6-311++G(2d,2p) basis set and the Truhlar group hybrid functional M0-62X [22], as this was demonstrated to be a viable compromise in terms of accuracy and computational cost [20]. All calculations were performed in vacuo, as we found in our earlier paper [20] that implicit solvation with water did not have any significant impact on the geometries or charges of the molecules.

### 2.1. Transannular Interactions in Protopine Analogues

Previous results [20] on protopine (**8**) and some hypothetical analogues (**9**) with R = CH_3_ are included in Table 1 for comparison purposes. The nitrogen is sp^3^ hybridised and substituted with a methyl group (substituent R in Figure 2 above) and various Y groups are employed. The table also presents new results for analogues where the nitrogen is substituted with a hydrogen atom and a hydroxyl group (R = H, OH). Appendix A provide further details on the compounds, such as geometrical parameters, relative energies if several conformations were investigated and orbital overlaps between the lone pair on the nitrogen and the closest Y group atom. Appendix A illustrates the geometrical parameters provided in the tables.

In all the protopine analogues (**9**) with an sp^3^ hybridised nitrogen and R = Me (see Appendix A for details), transannular interactions between N and Y had a significant influence on the preferred conformation of the 10-membered ring and hence the overall molecular shape, which was generally flat but with some buckling. The degree of the interaction, as indicated by the bond critical point (BCP) electron density between N and Y, varied from weak to strong depending on the nature of Y. For example, with Y being an sp^3^ hybridised carbon unit, only a weak interaction was predicted, but with Y = O or S it was moderate, and with SO, CS and CO it was relatively strong. In these last two cases an n-π* interaction [18] is likely to occur. With the sulfinyl group (sulfoxide) it is possible there is an n-S-based σ hole interaction in play based on other theoretical studies of the S-O bond in dimethyl sulfoxide [23]. With Y = CS, the interaction allowed formation of a single dative bond resulting in formal zwitterionic charge separation. The lowest energy form, however, was the one with the n-π* interaction (ΔE = 1.5 kcal/mol) [20]. A dative bond was also seen as the lowest energy species with Y = BOH through the nitrogen lone pair being donated into the boron vacant p orbital, as occurs in N^…^B intermolecular interactions.

The strength of the transannular interaction, as measured by the electron density between N and Y, is not sensitive to substitution of the nitrogen, which is unexpected. The minimum energy conformations for molecules (**9**) with R = Me were used to change the R group or atom, followed by energy optimization. Limited evaluation of other conformations and forms was performed for Y = CO and Y = CS. Appendix A provides full details. For Y = CS and R = H and R = OH, the forms with dative bonds are more stable than the ones with an n-π* interaction (ΔE = 0.75 kcal/mol for R = H and 1.44 kcal/mol for R = OH). For comparison purposes, the conformation with an n-π* interaction is provided in Table 1. The shape of the molecules is also not affected by the R group or atom. All molecules in Table 1 are flat with a step in the medium-sized ring and a small twist between the two aromatic rings (see Figure 5 below).

As expected, the electrostatic charge on the nitrogen is very sensitive to substitution of the N. It is always negative for R = H, and usually positive for R = CH_3_, except for Y = CS and Y = SO. For R = OH, the charge on N is similar to the charge for R = CH_3_, except for Y = BOH and Y = O. The charges on the Y group atoms are not affected very much by substitution on the nitrogen.

When mapping the electrostatic potential values onto electron density surfaces (Figure 3), the effect of the Y group dominates the surface potential. All molecules in Figure 3 are shown in similar orientations with the Y group (BOH for Figure 3a–c and CH_2_ for Figure 3d–f) situated in the middle of the lower part of the molecule, pointing forward. The OH group of Y = BOH gives rise to an extended area of negative (red) potential (Figure 3a–c). The effect of N substitution also becomes apparent with pronounced negative (red) potential around the OH substituent (also pointing forward in Figure 3c,f).

### 2.2. Conformational Preferencing and Resultant Molecular Shapes in Imino and Azo Group Embedded Rings

The results obtained with the same Y groups when the nitrogen in the medium-sized ring is sp^2^ hybridised are given in Table 2 (cyclic imino group) and Table 3 (cyclic azo group). Appendix A provide full details. Figure 4 illustrates the structures of the systems calculated. For Y = CO, several conformations were calculated and the conformation with the minimum energy has been identified in Table 2 and Table 3.

The model hypothetical imino and azo systems (**10**) and (**11**)**,** respectively, provided some interesting further insights into the N^…^Y transannular interactions. In these two systems the nitrogen is sp^2^ hybridised, but still with a nitrogen lone pair of electrons. The relative orientation of groups around the C=N and N=N bonds was explored and transannular interactions were only possible for imino or azo groups embedded in the ring in a *trans* disposition. The interactions occurred with most of the Y groups as indicated by the BCP electron densities including a single dative bond with Y = BOH. The electrostatic potentials at the N and ring Y atoms or groups were also reflective of these interactions.

As a result, the shapes of the preferred conformers were mainly flat with some twisting or a U shape with some twisting. A boomerang type bent shape was also seen in some cases.

While the medium sized ring system in (**10**) is hypothetical, it was considered to be present in the Poppy alkaloids fumaridine and fumaramine [24], but later work by Shamma and co-workers [25] revised the structural assignments to hydrastine imide and bicuculline imide, respectively. These compounds do not have a fused 10-membered ring system and belong to the phthalideisoquinoline alkaloid group.

The shapes referred to in Table 2 and Table 3 are illustrated in Figure 5 and it is interesting to note that contrary to expectations, the shapes of the trans imino and the trans azo compounds are similar to the compounds with an sp^3^ hybridised N. The ‘flat with a step’ shape also often incorporates various amounts of twist between the aromatic rings (Figure 5a,b), as does the more curved boomerang shape (Figure 5c) and the U shape (Figure 5d).

As was observed in the previous section, the electrostatic potentials on the surface are dominated by the Y group; this is illustrated for Y = BOH and Y = CS in Figure 6. The potentials for the imine and azo compounds are very similar and both are also similar to the protopine analogue.

### 2.3. Shape and Electrostatic Potential Comparisons

The calculated results for coptisine (**4**) and reduced derivatives (**5**) and (**6**) are given in Table 4 for comparison with the compounds modelled in this study. The strongest transannular interactions (densities of 0.11) are found for Y = BOH in all three compounds (**9**), (**10**) and (**11**), and they are indicative of a dative bond [26]. For the other Y groups, transannular interactions can be rationalized as n-π* orbital overlaps (Appendix A). In coptisine and the reduced derivatives, the densities reflect the covalent bonding in the central C–N linkage.

The shapes of the compounds with appreciable transannular interactions are mostly flat with a step and some degree of twisting, and thus mimic tetrahydrocoptisine very well and dihydrocoptisine to some extent. Coptisine is a flat lipophilic cation and none of the compounds investigated here mimic coptisine. The electrostatic potentials mapped onto the density surface for (**4**), (**5**) and (**6**) are presented in Figure 7.

### 2.4. Drug Design Implications

A number of implications for drug design have been revealed by these computational studies, in which it is clear that transannular or cross-ring interactions in N-based 10-membered ring systems can influence conformational preferencing (lowest energy conformations). This affects the overall molecular shape together with electrostatic potentials on the surface of the molecules and charges on the key atoms involved. This then could influence the way in which these small molecules may interact with individual biological targets. As an example, the alkaloid protopine (**8**) is known [27] to have in vitro antiplasmodial activity (Plasmodium falciparum) and is thus of interest as a potential antimalarial drug. While its mode of action is not known, in silico fishing and pharmacological profiling studies [28] have indicated that one of the biological targets for protopine in the antimalarial activity context may be the enzyme enoyl-acyl carrier reductase PfENR, which is involved in crucial fatty acid biosynthesis in Plasmodium falciparum. Analogues of protopine, as discussed in this paper, may thus show similar and potentially better and more selective binding activity to PfENR, particularly if the Y group was CS, SO, or BOH. With BOH there would also be potentially increased opportunities for both intermolecular H-bond donor or H-bond acceptor interactions with the target protein.

There is also great interest in protopine ((**8**) or (**9**), R = Me, Y = CO) in the anti-cancer drug area as it is able to selectively bind to key DNA-based targets. In this context, protopine has been shown to bind strongly with the G-quadruplex DNA in the human c-myb (myeloblastosis) promoter gene, and good selectivity for the G-quadruplex was seen over duplex DNA [29]. Protopine analogues with different Y groups (**9**), together with the cyclic imino (**10**) or azo (**11**) systems, could thus be of considerable interest in gaining even greater binding affinities while maintaining selectivity for the G-quadruplex motif.

In general, with the series of compounds (**9**), (**10**) and (**11**), there is scope for medicinal chemists to fine tune the electrostatic potential on the surface and the groups available for interactions with potential targets.

The compounds (**9**), (**10**) and (**11**) are also attractive targets for synthesis as they are predicted to be bioavailable after oral administration. With molecular weights (MW) below 400, calculated octanol/water partition coefficients (logP) below four, number of hydrogen bond donors (HBD) below five and hydrogen bond acceptors below 10 (2 times 5), the compounds meet the Lipinski rule of five [30]. The compounds also meet the Veber rules for oral bioavailability, with calculated topical polar surface areas (TPSA) well below 140 Å^2^ and fewer than 10 rotatable bonds [31].

## 3. Materials and Methods

### 3.1. Software

The Discovery Studio Visualizer (DS) is freely available for not-for-profit research and can be downloaded from the company website (https://discover.3ds.com/discovery-studio-visualizer-download; accessed on 25 January 2023). This software was used for structure editing and visualisation.

Gaussian 16 was used on the kunanyi HPC cluster of the Tasmanian Partnership for Advanced Computing (TPAC) at the University of Tasmania.

Gaussian 16W, the external NBO-6 utility and GaussView 6.0 were installed on a desktop PC of the School of Natural Sciences (Chemistry), College of Sciences and Engineering, University of Tasmania.

The predictions of molecular properties were performed using the molinspiration server (molinspiration.com; accessed on 25 January 2023).

### 3.2. Methods

A detailed description of the methods used, and examples of Gaussian Job Files can be found in a previous publication [20].

Briefly, all trial geometries were optimized in vacuo using an augmented 6-311++G(2d,2p) basis set and the Truhlar group hybrid functional M06-2X [22], as this was demonstrated to be a viable compromise in terms of accuracy and computational cost [20].

All reported structures are minima, i.e., there are no imaginary frequencies. The Gaussian 16 software was used for these calculations. The electrostatic potentials and electrostatic charges (POP=ESP) were calculated with the standard methods used in the Gaussian software. These are based on the PRISM algorithm by B. G. Johnson et al. [32].

NBO calculations were performed with Gaussian 16W, using the external NBO-6 utility. Topological analysis of the electron density was performed as an NBO calculation as described by F. Weinhold [33] using the NBCP keyword.

The molinspiration server uses a method based on group contributions for the prediction of octanol/water partition coefficients (logP). For the prediction of topical polar surface areas (TPSA), a method based on sums of fragments is used [34].

## 4. Conclusions

Density Functional Theory has value in indicating how ring nitrogen-based transannular interactions in actual (protopine) and related and novel model systems can influence key parameters for the design of new drugs, including conformational preferencing, molecular shapes, the electrostatic potentials on the surface and electrostatic potential charges on atoms or groups involved in interactions with biological targets.

Some key targets for subsequent synthesis and biological activity studies have been identified from this work, particularly where the lone pair of electrons on nitrogen can interact with Y groups across the 10-membered ring. Y groups with an sp^2^ hybridised carbon interacting with the nitrogen (CO and CS) and Y = BOH and SO are of particular interest, as they can form strong transannular interactions. In addition, model cyclic analogues involving an embedded azo group also have the potential for photo-switching control of bioactivity, as only the compounds with the N=N groups embedded in the ring in a *trans* disposition are capable of transannular interactions, leading to different preferred conformations for *cis* and *trans* forms.

## Figures and Tables

**Figure 1 molecules-28-01631-f001:**
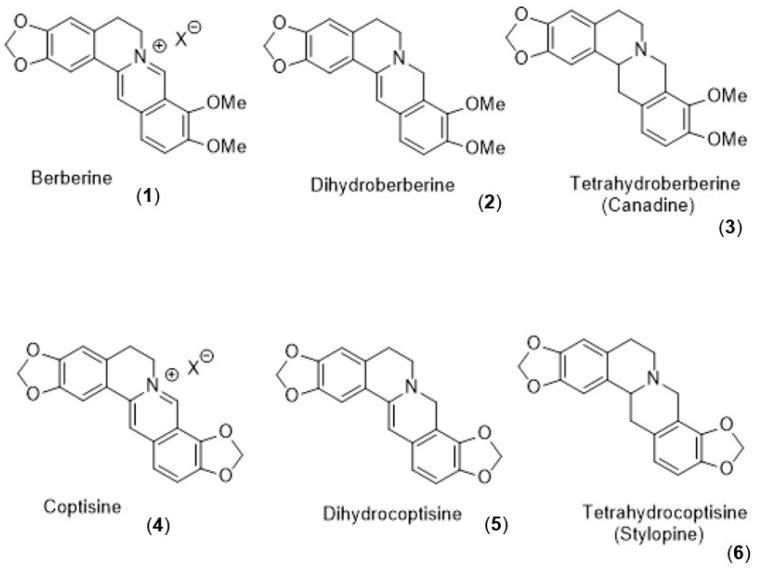
Structures of berberine and coptisine and their dihydro and tetrahydro reduced derivatives.

**Figure 2 molecules-28-01631-f002:**
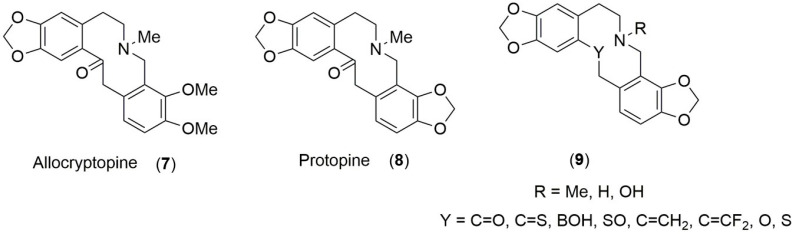
Structures of allocryptopine, protopine and related model compounds.

**Figure 3 molecules-28-01631-f003:**
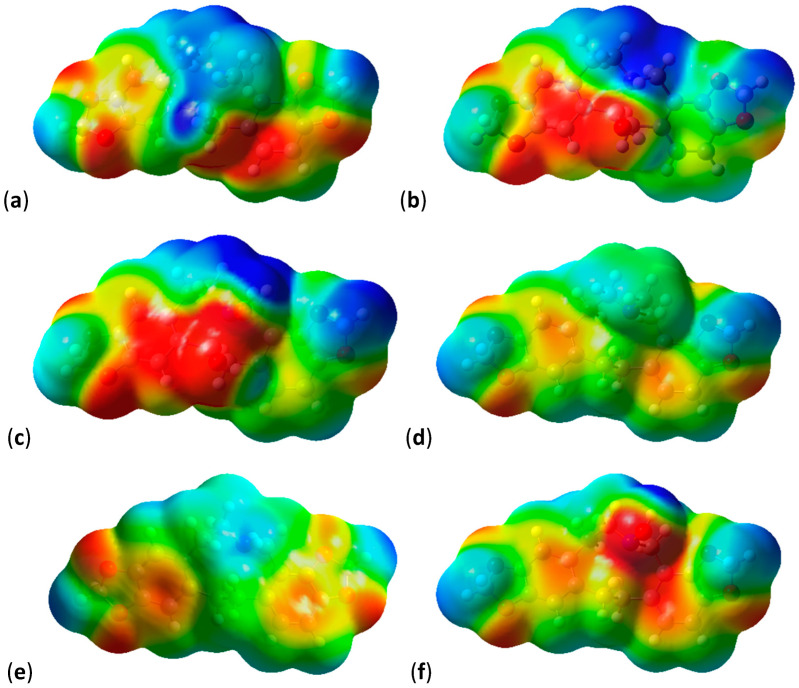
Electrostatic potentials mapped onto electron density surfaces of protopine analogues (**9**). Negative potentials are red, positive potentials are blue, scale 0.029. Densities at an isovalue of 0.0004 electrons/au^3^. (**a**) Y = BOH, R = CH_3_; (**b**) Y = BOH, R = H; (**c**) Y = BOH, R = OH; (**d**) Y = CH_2_, R = CH_3_; (**e**) Y = CH_2_, R = H; (**f**) Y = CH_2_, R = OH.

**Figure 4 molecules-28-01631-f004:**
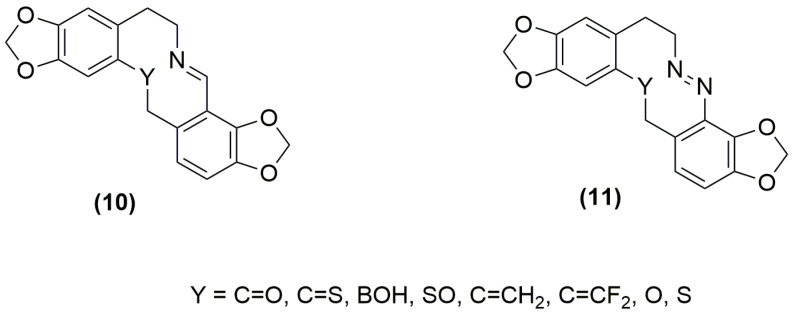
Structures of model imino- and azo-group embedded systems.

**Figure 5 molecules-28-01631-f005:**
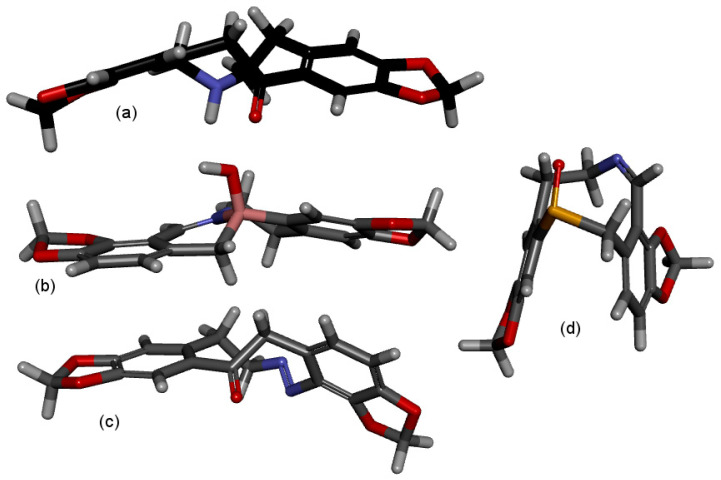
Examples of common shapes. (**a**): flat with step and twist; (**9**), Y = CO, R = H. (**b**): flat with step; (**10**), trans-imine, Y = BOH. (**c**): boomerang; (**11**), trans-azo, Y = CO. (**d**): U with twist; cis-imine, Y = SO.

**Figure 6 molecules-28-01631-f006:**
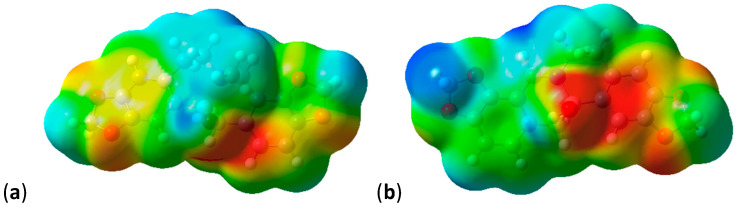
Electrostatic potentials mapped onto electron density surfaces of compounds with Y = BOH and Y = CS. Negative potentials are red, positive potentials are blue, scale 0.040. Densities at an isovalue of 0.0004 electrons/au^3^. (**a**) Y = BOH, protopine analogue (**9**); (**b**) Y = BOH, trans imine (**10**); (**c**) Y = BOH, trans azo (**11**); (**d**) Y = CS, protopine analogue (**9**); (**e**) Y = CS, trans imine (**10**); (**f**) Y = CS, trans azo (**11**).

**Figure 7 molecules-28-01631-f007:**
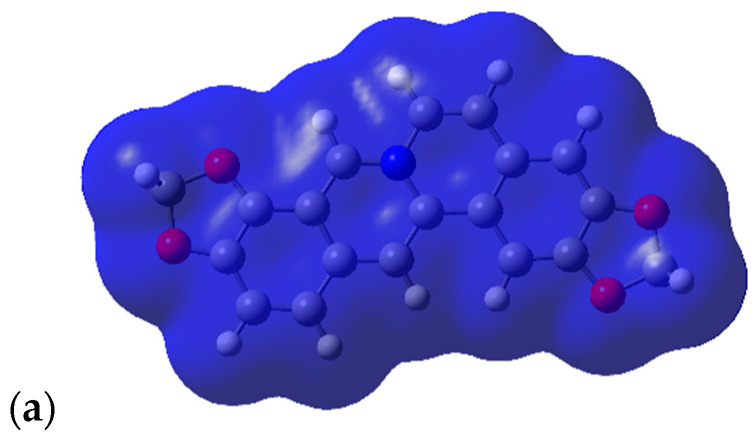
Electrostatic potentials mapped onto electron density surfaces of coptisine analogues. Negative potentials are red, positive potentials are blue, scale 0.034. Densities at an isovalue of 0.0004 electrons/au^3^. (**a**) Coptisine; (**b**) dihydrocoptisine; (**c**) tetrahydrocoptisine.

**Table 1 molecules-28-01631-t001:** Calculated electrostatic charges and electron densities for protopine (**8**) and model analogues (**9**) with an N-R group.

Y,R	Charge N	Charge Y1 ^1^	Charge Y2 ^2^	Density ^3^
CO,CH_3_	0.104	0.835	−0.579	0.023
CO,H	−0.396	0.790	−0.555	0.019
CO,OH	0.168	0.870	−0.534	0.018
BOH,CH_3_	0.348	0.910	−0.891	0.109
BOH,H	−0.240	0.805	−0.806	0.098
BOH,OH	−0.059	0.834	−0.730	0.073
CS,CH_3_	−0.042	0.248	−0.293	0.021
CS,H	−0.345	0.211	−0.280	0.019
CS,OH	0.053	0.250	−0.229	0.019
SO,CH_3_	−0.027	0.366	−0.523	0.022
SO,H	−0.416	0.297	−0.494	0.021
SO,OH	−0.041	0.475	−0.540	0.022
S,CH_3_	0.132	−0.202	NA	0.015
S,H	−0.265	−0.220	NA	0.014
S,OH	0.180	−0.135	NA	0.014
O,CH_3_	0.019	−0.254	NA	0.014
O,H	−0.521	−0.281	NA	0.014
O,OH	0.156	−0.235	NA	0.015
CH_2_,CH_3_	0.204	0.287	−0.010, −0.103	0.019 (N^…^H)
CH_2_,H	−0.321	0.314	−0.023, −0.092	0.018 (N^…^H)
CH_2_,OH	0.233	0.167	−0.009, −0.009	0.019 (N^…^H)

^1^: atom of Y group closest to the N; ^2^: second atom of Y group away from the N, if present; ^3^: electron density between N and Y1 (atomic units); NA: not applicable.

**Table 2 molecules-28-01631-t002:** Calculated electrostatic charges, electron densities and shapes for model ring-embedded imino group analogues (**10**).

Y ^1^	Charge N	Charge Y1 ^2^	Charge Y2 ^3^	Density ^4^	Shape
CO trans min	−0.420	0.712	−0.529	0.016	U with twist
CO trans	−0.321	0.787	−0.534	0.023	flat with step
CO trans2	−0.430	0.764	−0.519	0.016	boomerang
CO cis	−0.674	0.820	−0.512	none	boomerang
CO cis pinched	−0.598	0.607	−0.494	none	U with twist
BOH trans	−0.253	0.585	−0.746	0.112	flat with step
BOH cis	−0.648	0.600	−0.689	none	U with twist
CS trans	−0.251	0.271	−0.275	0.022	flat with step
CS cis	−0.522	0.165	−0.193	none	U with twist
SO cis	−0.564	0.137	−0.408	none	U with twist
SO trans	−0.265	0.326	−0.511	0.031	flat with step
S trans	−0.330	−0.164	NA	0.015	boomerang
O trans	−0.409	−0.225	NA	0.017	flat with step
CH_2_ trans	−0.291	0.237	−0.035/−0.013	0.019 (N^…^H)	boomerang
CF_2_ trans	−0.250	0.747	−0.267/−0.290	0.019 0.014 (N^…^F)	boomerang
CCH_2_ trans	−0.279	0.361	−0.628	0.017	flat with step
CCF_2_ trans	−0.280	−0.012	0.279	0.018	flat with step

^1^: cis, trans refer to the orientation of groups around the C=N bond; ^2^: atom of Y group closest to the N; ^3^: second atom of Y group away from the N, if present; ^4^: electron density between N and Y1 (atomic units); NA: not applicable.

**Table 3 molecules-28-01631-t003:** Calculated electrostatic charges, electron densities and shapes for model ring-embedded azo group analogues (**11**).

Y ^1^	Charge N	Charge Y1 ^2^	Charge Y2 ^3^	Density ^4^	Shape
CO trans	−0.009	0.584	−0.497	0.019 (N33)	boomerang
CO trans2	−0.154	0.729	−0.477	0.016 (N33)	boomerang
CO trans min	−0.089	0.534	−0.481	0.016 (N33)	U with twist
CO cis	−0187	0.553	−0.486	none	U with twist
CO cis2	−0.145	0.635	−0.470	none	U with twist
BOH trans	0.225	0.896	−0.878	0.108 (N32)	flat with step
BOH cis	−0.158	0.617	−0.656	none	U with twist
CS trans	0.039	0.081	−0.239	0.018 (N33)	flat with twist
CS cis	−0.192	0.086	−0.150	none	U with twist
SO trans	−0.045	0.246	−0.469	0.023	flat with step
SO cis	−0.185	0.214	−0.467	none	U with twist

^1^: cis, trans refer to the orientation of groups around the N=N bond; ^2^: atom of Y group closest to the N; ^3^: second atom of Y group away from the N, if present; ^4^: electron density between N and Y1 (atomic units).

**Table 4 molecules-28-01631-t004:** Calculated electrostatic charges, shape and electron densities for coptisine (**4**) and reduced derivatives (**5**) and (**6**) with a central C-N linkage.

Compound	Charge N	Charge C	Charge H ^1^	Density ^2^	Shape
Coptisine	0.262	0.230	NA	0.288	Flat
Dihydrocoptisine	−0.204	0.334	NA	0.296	Flat with twist
Tetrahydrocoptisine	−0.405	0.389	0.033	0.265	Flat with step

^1^: H on C of central linkage; ^2^: electron density between N and C (atomic units); NA: not applicable.

## Data Availability

Not applicable.

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
