# Peer review of "Computational Evaluation of N-Based Transannular Interactions in Some Model Fused Medium-Sized Heterocyclic Systems and Implications for Drug Design"

_molecules, 2023, doi:10.3390/molecules28041631_

Round 1

Reviewer 1 Report

The manuscript ID: molecules-2092984 entitled “Computational Evaluation of N-based Transannular Interac-tions in Some Model Fused Medium-Sized Heterocyclic Systems and Implications for Drug Design” is a continuation of the Authors' previous works on interactions in model heterocyclic systems containing the N atom and fully deserves publication in the Molecules.

I can recommend this manuscript for publication in the present form but I think that the distributions of the electrostatic potential presented in the Figures 3, 6 and 7 would be clearer if a different visualization technique with a visible skeleton of the molecule was used.

Author Response

We take the point that it would be good to show the molecule skeletons in Figures 3, 6 and 7. This has now been done with the molecules visible under a semi-transparent surface.

Reviewer 2 Report

Minor revision is required. Several concerns and comments are below:

1. The definitions and the computational methods of electrostatic charges and electron densities should be provided. Different population schemes would lead to different charges and densities. 

2. The "NA" values in Tables 1~2 and 4 should be explained.

3. In Figures 6-7, the electrostatic potentials of different compunds are dispalyed with different scales. It is unobjective.

4. M062X should be "M06-2X"

Author Response

Point 1: The computational methods with regard to the electrostatic charges and electron densities have now been provided.

Point 2: NA means "not applicable". This has been added to the footnotes of the relevant tables.

Point 3: Thank you for this. The electrostatic potentials of the different compounds in Figures 3, 6 and 7 are now displayed with the same scales in each figure.

Point 4: This correction has now been made, thanks.

Reviewer 3 Report

The article presents a computational analysis of N-based transannular interactions in protopine analogues, and derivatives containing imino or azo groups. These are model fused medium-sized heterocyclic systems which are analyzed as potential drugs. The interactions N…Y are evaluated with several groups: CO, CS, BOH, SO, CCH2, CCF2, O, S, and CH2. With Y= CO, CS, BOH and SO were observed strong interactions. These interactions are relevant because they can influence in the conformational preferences, in the shape, and in the electron distributions at key reactive sites for biological target interactions. The prediction of the strong transannular interactions could be useful in the synthesis and design of new compounds.

In general, the article contains original and interesting information which justifies the relevance and scientific motivation of the study. However, the manuscript has several improvements to be published in the Molecules journal. The methods do not consider the solvent effect, which is important to analyze in the prediction of transannular interactions, especially when molecules are analyzed for medical chemist and drug design. In addition, some discussions should be improved for better understanding for the reader. Therefore, my recommendation is Major Revision. I suggest several changes and improvements as follows: 

Major revisions:

In Abstract section

Page 1, lines 15 and 16, what general characteristics should be the atoms o groups Y to carry out the adequate N…Y interactions? It should be mentioned for better understanding of the reader.

In Introduction section

Page 3, lines 69-73, the paragraph: “The computational evaluation has also been taken further to include ring-embedded imino (10) [9, pp. 79-81] or azo (11) group nitrogens and atoms or groups Y across the 10-membered ring system, the aim of which was to assess whether some flattening of the ring could be achieved while maintaining additional control of conformational preferencing [21] through transannular N…Y interactions.” should be rewritten in short sentences for better clarity for the reader.

Page 3, lines 69 and 70, when authors mention: “ring-embedded imino (10) [9, pp. 79-81] or azo (11) group nitrogens”, they should mention that the compounds (10) and (11) are shown in the Figure 4.

In general, the redaction of the Introduction section should be improved. Short and clear sentences should be used rather than large phrases with a lot of ideas together.

In Results and Discussion section

Page 3, lines 86-89, the paragraph: “The DFT-based computations were performed as described earlier using an augmented 6-311++G(2d,2p) basis set and the Truhlar group hybrid functional M062X [22] as this was demonstrated to be a viable compromise in terms of accuracy and  computational cost [20].” should be removed in this part. This information is already included in the Materials and Methods section.

Page 3, lines 105-110, the following phrases: “For example, with Y being an sp3 hybridised carbon unit, only a weak interaction was predicted, but with Y=O or S it was moderate and with SO, CS and CO it was relatively strong. In these last two cases an n-π* interaction [18] is likely to occur and with the sulfinyl group (sulfoxide) it is possible there is an n-S-based σ hole interaction in play based on other theoretical studies of the S-O bond in dimethyl sulfoxide [23].” should be rewritten for better clarity for the reader.

Page 4, lines 133-135, in Table 1, it is not clear which is the Charge Y2, one footnote in the table is necessary for explaining it.

Page 5, lines 140-142, page 8, lines 206-208, and page 9, lines 233-235, in Figures 3, 6 and 7, I suggest that the electrostatic potential mapped on the electron density surfaces be transparent instead of solid surfaces to be able to visualize the atoms involved in the transannular N…Y interactions.

Page 2, Figures 1 and 2, how many new molecules are studied? Berberine (1), dihydroberverine (2), tetrahydroberberine (3), coptisine (4), dihydrocoptisine (5), tetrahydrocoptisine (6), allocryptopine (7), protopine (8), and some derivatives of protopine (9) are included in the study of the reference [20] (B. Bremner and R. Griffith, Comput. Theor. Chem., 2022, 1208, 113543). In these molecules are again included in the current study for sake of comparison or what contribution they do to this work? This should be well justified by the authors please.

In Materials and Methods section

Page 10, line 277, what software was used to calculate the electrostatic charges and electron densities between N and Y? It should be clearly mentioned in this section.

Page 10, line 293, why the solvent effect was not included in the calculations? It would be very interesting to analyze the solvent effect on the density parameters in comparison with in vacuo calculations. I suggest including the solvent effect in aqueous solution.

In Conclusions section

Page 11, lines 305-314, the conclusions should be improved from the suggestions carried out.

Author Response

Point 1 about Abstract: This (general characteristics of Y groups forming the interaction) was already mentioned in lines 18 and 19 and the text there has been altered slightly.

Point 2 about rewriting lines 69 to 73: We think this writing is clear and do not wish to make a change.

Point 3 about mentioning Figure 4 on page 3: This has been done, thank you.

Point 4 about writing of Introduction: We think the sentences are clear as is and to cut down to a number of short sentences would adversely affect the flow of the presentation.

Point 5 about removing details of computations: It is common practice to briefly summarise what was done, and why, at the beginning of the Results section. We believe this is helpful for clarity.

Point 6 about rewriting lines 105-110: We think this is clear as is and do not wish to change our style of writing.

Point 7: A footnote has been added to Tables with Y2 to explain the meaning of Y2.

Point 8: We take the point that it would be good to show the molecule skeletons as well. This has now been done with the molecules visible under a semi-transparent surface.

Point 9: More explanation/justification has been added in the text on page 2 with regard to the new molecules studied and why some previous results are included for necessary comparison purposes.

Point 10: The software and methods used to calculate the electrostatic charges and electron densities has been described.

Point 11 about solvent effects: This is a reasonable point. However, we did some calculations in implicit solvent as presented in our first paper (reference 20) but found that this makes not very much difference at all. Hence this was not done with any of the new compounds in this current paper. A comment to this effect has now been added to the manuscript.

Point 12: We think the conclusions are clearly and adequately presented and don't think change is necessary.

Reviewer 4 Report

Authors presented in this current manuscript is duplication of your previous paper ""Bremner, J.; Griffith, R. Density Functional Theory Assessment of Transannular N…Y Interactions in Some Medium-Sized Heterocycles. Comput. Theor. Chem. 2022, 1208, 113543""". So, I cannot accept in current results using same and similar compounds. Authors needs to investigate new things, not the same and similar compounds study. Authors clearly states in the current manuscript what is the exact difference between your previous CompTheorChem'22 results. I dont think except DFT method and basis set difference between previous published and current manuscript.

Author Response

We do not agree with the reviewer's assessment about the lack of novelty of this manuscript.

The cyclic imines and azo compounds are not "similar compounds"; they are fundamentally very different to what was done before. With the sp3 hybridised nitrogen being replaced with an sp2 hybridised one, we were definitely investigating “new things”. None of these compounds have been synthesised, or investigated computationally.

Additionally, a number of significant new variations have been investigated in this current work involving different groups R (with different electronic and steric characteristics) on the sp3 hybridised nitrogen in order to probe further the impact on transannular interactions and conformations.

We believe we have made the difference between the current manuscript and our previous paper very clear in the revised manuscript.

Round 2

Reviewer 3 Report

I have carefully examined the revised version of the article: Computational Evaluation of N-based Transannular Interactions in Some Model Fused Medium-Sized Heterocyclic Systems and Implications for Drug Design (molecules-2092984).

The authors made efforts on solving some concerns from the reviewers and the manuscript was improved to be published in the Molecules journal. Some suggestions were taken into account and it is clearer that the manuscript is a continuation of the previous works by the authors and several compounds were again included for comparison purposes. Also, the conclusions were slightly improved. Therefore, my recommendation is Accept in present form.

Reviewer 4 Report

Accept after minor revisions. Still Can be improved NBO analysis like Donor-Acceptor Interactions like E2 (second orbital interaction energies).